# “Attractive” Treatment for Abdominal Aortic Aneurysm Repair: Magnetic Localization of Silk-Iron Packaged Extracellular Vesicles

**DOI:** 10.3390/jfb16110395

**Published:** 2025-10-22

**Authors:** Ande X. Marini, Kiran J. McLoughlin, Amanda R. Pellegrino, Golnaz N. Tomaraei, Bo Li, John A. Curci, Mostafa Bedewy, Justin S. Weinbaum, David A. Vorp

**Affiliations:** 1Department of Bioengineering, University of Pittsburgh, Pittsburgh, PA 15261, USA; ande.marini@pitt.edu (A.X.M.); kiran_mc@pitt.edu (K.J.M.); amp370@pitt.edu (A.R.P.); JUW51@pitt.edu (J.S.W.); 2Department of Industrial Engineering, University of Pittsburgh, Pittsburgh, PA 15261, USA; GON2@pitt.edu (G.N.T.); mbedewy@pitt.edu (M.B.); 3Department of Vascular Surgery, Vanderbilt University, Nashville, TN 37232, USA; bo.li.1@vumc.org (B.L.); john.curci@vumc.org (J.A.C.); 4Department of Surgery, Veterans Affairs Tennessee Valley Healthcare System, Nashville, TN 37212, USA; 5Department of Graduate Studies, Meharry Medical College, Nashville, TN 37208, USA; 6Department of Mechanical Engineering and Materials Science, University of Pittsburgh, Pittsburgh, PA 15261, USA; 7Department of Chemical and Petroleum Engineering, University of Pittsburgh, Pittsburgh, PA 15261, USA; 8McGowan Institute for Regenerative Medicine, University of Pittsburgh, Pittsburgh, PA 15219, USA; 9Department of Pathology, University of Pittsburgh, Pittsburgh, PA 15261, USA; 10Department of Surgery, University of Pittsburgh, Pittsburgh, PA 15213, USA; 11Department of Cardiothoracic Surgery, University of Pittsburgh, Pittsburgh, PA 15213, USA; 12Clinical & Translational Sciences Institute, University of Pittsburgh, Pittsburgh, PA 15213, USA; 13Magee Women’s Research Institute, University of Pittsburgh, Pittsburgh, PA 15213, USA

**Keywords:** abdominal aortic aneurysm, extracellular vesicles, drug delivery, regenerative medicine, silk microparticles, extracellular matrix, degradation

## Abstract

Abdominal aortic aneurysm (AAA) is a dilatation of the distal aorta to a diameter of 50% or more of its normal size of about 2 cm. Risk of aortic rupture can be nearly eliminated with either open surgery or endovascular repair. Procedural risks limit the value of these interventions unless the diameter of the aneurysm has reached a critical threshold (established as 5.5 cm in men or 5.0 cm in women). Thus, patients are monitored until this threshold is reached. Approximately 80% of small AAA will grow and exceed the threshold, providing a therapeutic window for altering this natural history and reducing the risk of rupture. Previous work in our lab has utilized adipose-derived mesenchymal stem cells (ASCs) to treat AAA in vivo, preserving elastic fibers and slowing aneurysm expansion. This work sought to create a delivery system for therapeutic extracellular vesicles (ASC-EVs) secreted by ASCs. Our delivery system incorporated the biocompatibility of regenerated silk fibroin (RSF), the magnetic moveability of iron oxide nanoparticles (IONPs), and the regenerative nature of ASC-EVs to create silk-iron packaged extracellular vesicles (SIPEs). Using this system, we tested the ability to magnetically localize the SIPEs and release their encapsulated ASC-EVs to exert their regenerative effects in vitro. We were successful in magnetically localizing the SIPEs in vitro and silk-iron microparticles (SIMPs) in vivo and in detecting their releasates via flow cytometry and cellular uptake assays. However, while their releasates were detected, their biological effects were diminished compared to unencapsulated controls. Thus, additional optimization related to loading efficiency is needed.

## 1. Introduction

Cardiovascular disease is the leading cause of death in the United States [1], with abdominal aortic aneurysm (AAA) alone being the 13th highest cause [2,3]. AAA is a life-threatening dilation and structural compromise of the distal aorta wall. This aortic expansion is caused by vascular smooth muscle cell (SMC) dysfunction, degradation of the extracellular matrix, and unregulated inflammation. Adipose-derived mesenchymal stem cells (ASCs) have shown promise for the treatment of AAA by our lab [4] and by others [5,6]. The ARREST trial was the first clinical trial to utilize ASCs for the treatment of AAA [7]. Both our lab [8] and others [9] have utilized autologous ASCs for other applications [10], and both have found deficiencies with autologous ASCs from at-risk populations. Additionally, there are potential regulatory hurdles associated with cell therapy, as ASCs could engraft [11] and differentiate into a variety of unwanted cell types or form teratomas [12]. In recent years, ASCs have been utilized for their paracrine effects [13,14,15,16,17], as they secrete cytokines and other factors that can have regenerative effects. Thus, extracellular vesicles (ASC-EVs) and a portion from this secretome have been used in recent publications to promote regeneration [18,19,20,21,22,23,24,25,26,27,28,29,30,31].

The small size of ASC-EVs allows them to penetrate a variety of tissues, but this small size also allows them to be easily cleared [32]. Thus, systemic delivery of ASC-EVs requires high dosages to account for this clearance. These ASC-EVs then can be found in the spleen, liver, gastrointestinal tract, lungs, and kidneys [33,34,35] when they are cleared, potentially leading to off-target effects. Creating a localizable system for ASC-EV release could alleviate this issue. Previously, we have shown that microparticles composed of regenerated silk fibroin (RSF) and chemically conjugated with iron oxide nanoparticles (IONPs) can be magnetically localized (named silk-iron microparticles (SIMPs)) [36], and others have loaded cargo within silk microparticles [37,38,39,40,41,42,43]. Silk was chosen as the microparticle material for its biocompatibility [44], tunable properties, and versatile applications [45,46,47] in regenerative medicine; IONPs were chosen to add magnetic properties to the silk microparticles. ASC-EVs could then be incorporated during the microparticle fabrication process to surround the ASC-EVs with the IONPs-GSH + RSF mixture. Encapsulating ASC-EVs within SIMPs to create silk-iron packaged extracellular vesicles (SIPEs) could potentially combine the regenerative properties of silk, controlled release aspect of the microparticles, magnetic properties of iron, and therapeutic capacity of ASC-EVs.

In this study, our goals were to incorporate ASC-EVs into our silk-iron microparticle carriers for release and eventual uptake into cells, examine retention of their biological activity, assess their ability to induce extracellular matrix (ECM) production, determine their effects on degradative activity, and investigate whether the microparticles could be localized in an in vivo model of AAA. We sought to demonstrate that ASC-EVs remained within the SIPEs (analyzed via magnetic separation); they were then released (determined with flow cytometry) and subsequently taken up by SMCs (confirmed with uptake assays). SIPEs were also tested for their effects on ECM-related genes, ECM deposition and degradation, migration and proliferation of SMCs, and their ability to be localized within a rat model of AAA.

## 2. Materials and Methods

### 2.1. Cell Culture

RoosterBio adipose-derived mesenchymal stem cells (ASCs, RoosterBio Inc., Frederick, MD, USA, C46001AD, Lot: 210323,) at passage 4 were plated in a T175 flask in 20 mL of growth media (RoosterBio Inc., Frederick, MD, USA, RoosterNourish, SU-022 supplemented with RoosterBio Inc., Frederick, MD, USA, RoosterBooster SU-003). The media was replaced the next day to remove residual dimethyl sulfoxide. After 3 days of culture, cells were passaged into two 5-layer tower flasks (175 cm^2^ per layer), 3 million cells per tower flask. Cells were cultured in 75 mL of growth media in the tower flasks for 3 days. At >80% confluence, cells were washed twice with PBS. Then 110 mL of harvest media (RoosterBio Inc., Frederick, MD, USA, RoosterCollect, M2001) was added to each tower flask. Cells were cultured in RoosterCollect for 48 h to be used for extracellular vesicle isolation. Human aortic vascular SMCs from Cell Applications Inc. (San Diego, CA, USA, 354-05a) were cultured in supplemented basal media (SBM) (Cell Applications Inc, San Diego, CA, USA, 311K-500) between passages 6 and 9. The media was changed every 2–3 days. U937 cells from American Type Culture Collection (ATCC, Manassas, VA, USA, CRL-1593.2, Lot: 70057944) were cultured in phenol-red-free Roswell Park Memorial Institute media (RPMI, Gibco, Gaithersburg, MD, USA, 11835030) supplemented with 10% fetal bovine serum (FBS, Atlanta Biologics, Flowery Branch, GA, USA, S11550) and 1% penicillin/streptomycin (pen/strep, Gibco, Gaithersburg, MD, USA, 15070063).

### 2.2. Extracellular Vesicle Isolation or Preparation

ASC-EVs were obtained by one of three different methods: isolation via ultracentrifugation (UC), isolation via tangential flow filtration (TFF), or commercial purchase. UC isolation was performed as described previously [19,20], and TFF isolation was performed with a 300 kDa hollow fiber membrane filter, with EVs isolated in the retentate. Commercial EVs were directly sourced from RoosterBio Inc. (Frederick, MD, USA, RoosterVial Exosomes, E42059AD, Lot: EA20240002). EVs were counted with nanoparticle tracking analysis (NTA) and diluted with Dulbecco’s phosphate-buffered saline (DPBS) to achieve either 1 × 10^9^ particles/mL or 3 × 10^9^ particles/mL. Additional details can be found in the Appendix A.

### 2.3. Generation of Silk, Fabrication of IONPs, and Incorporation of Iron Oxide Nanoparticles (IONPs) into Regenerated Silk Fibroin (RSF) Solution

A common method of RSF synthesis was used [48]. Briefly, silk cocoons (Ayara) were degummed in 0.05 M Na_2_CO_3_ followed by a LiBr boil (9.6 M at 65 °C, 3.5 h). The solution was dialyzed over three days to remove LiBr and then filtered using cleanroom wipes (Contec™, Spartanburg, SC, USA, AMSI0001). RSF was stored at 4 °C until use. IONP fabrication and IONPs and RSF incorporation were performed as described previously [36]. To fabricate IONPs, 0.05 M FeCl_2_·4H_2_O (Millipore Sigma, St. Louis, MO, USA, 220299, 98% reagent grade) and 0.1 M FeCl_3·_6H_2_O (Millipore Sigma, St. Louis, MO, USA, F2877, 98% reagent grade) were combined in 100 mL of ultrapure water in a round-bottom flask under argon (Ar) flow. Then, 100 mL of NH_4_OH was added dropwise, and the solution was stirred at 550 rpm for 30 min at room temperature and then 30 min at 70 °C. The solution was then cooled under Ar flow. IONPs were then washed with ultrapure water and ethanol and then dried overnight before storage in a desiccator. Conjugation of the IONPs with glutathione (GSH) used 16 mg of IONPs to 13 mg of GSH (Millipore Sigma, St. Louis, MO, USA, PHR1359) First, IONPs were mixed with 480 μL of ultrapure water and 160 μL of methanol and then sonicated to mix. Then GSH was added to the mixture and sonicated for 2.5 h with periodic agitation to ensure uniform sonication. The IONPs-GSH were then washed with ultrapure water and methanol, followed by a drying step before being stored in a desiccator. To mix the RSF with the IONPs-GSH, the IONPs-GSH were diluted with ultrapure water and sonicated for 20 min. Then, the RSF solution was added and sonicated for 90 s for uniform distribution. The final RSF concentration was reduced to 10 mg/mL to reduce over-precipitation of the RSF in buffer solution. A schematic of the fabrication and incorporation details can be found in Appendix A.

### 2.4. Fabrication of SIPEs and SIMPs

To fabricate SIPEs and SIMPs, the silk solution (10 mg/mL) with IONPs-GSH (5 mg/mL) (40 µL) was mixed with potassium phosphate buffer with ASC-EVs (either 1 × 10^9^ particles/mL or 3 × 10^9^ particles/mL, SIPEs) or without ASC-EVs (SIMPs) (2 M, pH = 8.0, 200 µL) and incubated at 37 °C overnight to form microparticles via a salting out method [36,37]. Following incubation, SIPEs/SIMPs were centrifuged at 10,000× *g* and 4 °C for 20 min. The supernatant was removed, and the SIPEs/SIMPs were resuspended in 200 µL of ultrapure water. The SIPEs/SIMPs were subsequently washed 5 times by centrifuging at the same speed and resuspending in the same volume of ultrapure water. After the final wash, SIPEs/SIMPs were resuspended in DPBS. A schematic of the fabrication of the SIPEs/SIMPs can be found in **Appendix A**.

### 2.5. Extracellular Vesicle Characterization

In accordance with the MISEV guidelines [49,50], ASC-EVs purchased from RoosterBio Inc. (Frederick, MD, USA) were characterized for size, concentration, protein concentration, and surface markers. Specific methods can be found in the Appendix A.

### 2.6. Magnetic Moveability Analysis with a Ninhydrin Assay for Protein Detection

Magnetic moveability with SIPEs and silk EV microparticles (silk EV MPs) was evaluated alongside SIMPs and silk microparticles (silk MPs). ASC-EVs used for SIPE and silk EV fabrication for this assay were purchased from RoosterBio Inc., Frederick, MD, USA, and a concentration of 1 × 10^9^ particles/mL of ASC-EVs was used. Following fabrication and washing, 100 µL of SIPEs, silk EV MPs, silk MPs or SIMPs were magnetically separated using a MagniSort (ThermoFisher Scientific, Pittsburgh, PA, USA, MAG-4902). Briefly, SIPEs/silk EV MPs/SIMPs/silk MPs were placed on the MagniSort for 3 min, and the supernatant (non-magnetic fraction) was removed carefully. The residue (magnetic fraction) was resuspended in 100 µL of ultrapure water. Magnetic and non-magnetic fractions were added to glass vials and boiled in 500 µL of 6 N HCl for 24 h at 110 °C. Then the vials were dried and resuspended in 500 µL of ultrapure water. The samples were then diluted 1:50 (4 µL into 196 µL of ultrapure water) and processed using ninhydrin reagent. Samples were run in triplicate. A standard curve with HCl boil-processed BSA was used to assess the protein amount of each fraction.

### 2.7. Flow Cytometry Releasate Experiments

ASC-EV release from SIPEs/SIMPs was enumerated using Exosome-Human CD63 Isolation/Detection Reagent Dynabeads™ (ThermoFisher Scientific, Pittsburgh, PA, USA, 10606D) according to the manufacturer’s instructions. CD63 is a known marker for ASC-EVs and thus was chosen to determine EV release. ASC-EVs (3 × 10^9^ particles/mL) used for SIPE fabrication for this assay were purchased from RoosterBio Inc., Frederick, MD, USA. Briefly, SIPEs/SIMPs were resuspended in 500 µL PBS and were placed onto a tube rotator in a 37 °C incubator for 7 days. Every 24 h, SIPEs/SIMPs were pelleted at 12,000× *g* for 20 min at 4 °C in a benchtop microcentrifuge, and the supernatant was collected and frozen at −80 °C. This supernatant was referred to as the releasate. After 7 days of collection, the releasates were thawed and processed. Then 100 µL of beads were washed with 1mL PBS, before discarding the supernatant and adding 500 µL of releasate and mixing, followed by incubation overnight at 4 °C.

The following day, 300 µL of PBS was added to the beads to wash unbound material, and the tubes were gently mixed before placing into a MagniSort™ Magnet (ThermoFisher Scientific, Pittsburgh, PA, USA, MAG-4902) for 1 min and discarding the supernatant. This was repeated with 400 µL PBS before resuspending bead-bound EVs in 2 × 100 µL flow cytometry staining buffer (FACS buffer) (ThermoFisher Scientific, Pittsburgh, PA, USA, 00-4222-26) across two 5 mL Falcon™ round-bottom polystyrene test tubes (ThermoFisher Scientific, Pittsburgh, PA, USA, 14-959-5). Then 100 µL of releasate was stained with 5 µL of anti-CD63-PE antibody (ThermoFisher Scientific, Pittsburgh, PA, USA, 12-0639-41) for 60 min at room temperature on an orbital shaker, protected from the light. The remaining 100 µL of bead-bound EVs was used as an unstained control to calibrate the flow cytometer. After staining, bead-bound EVs were washed with 300 µL of FACS buffer and gently mixed, before placing on the magnet for 1 min and discarding the supernatant. Bead-bound EVs were resuspended in 100uL FACS buffer before running on a MACSQuant^®^ Analyzer 16 Flow Cytometer (Miltenyi Biotec, Gaithersburg, MD, USA, 130-109-803). Then 20,000 events were collected per sample tube and analyzed using FlowJo v10.10 (BD Biosciences, Milpitas, CA, USA). Data was presented as percentage-positive CD63 events with the unstained control data used to set baseline fluorescence.

### 2.8. EV Uptake

Fibrin gels were seeded with SMCs as previously described [20,51]. Bovine fibrinogen type 1 (2 mL, 3.7 mg/mL, Millipore Sigma, St. Louis, MO, USA, #8630,) and 0.21 U/mL bovine thrombin (Millipore Sigma, St. Louis, MO, USA, 500 µL, #T7513) were mixed with 500 µL of SMCs (3 million cells/mL) resuspended in DMEM + GlutaMAX media (Gibco, Gaithersburg, MD, USA, 10564-011). To create the SMC-seeded constructs, 200 µL of the gel mixture was added to heat-stamped circular molds created with 7.94 mm (5/16”) diameter cork borers imprinted in 24-well plates. The final concentration of SMCs seeded in the gels was 5 × 10^5^ cells/mL. Gels were allowed to polymerize for 10 min at room temperature for 45 min at 37 °C before adding 1 mL of supplemented basal media with 12 mM ε-aminocaproic acid. Cells were incubated overnight in SBM to compact the gels before SIPEs were added in media. ASC-EVs isolated with TFF were used for this assay; purchased ASC-EVs were also used to validate that they could be encapsulated and released. For staining ASC-EVs, 2500 µL at 1 × 10^9^ particles/mL was stained with CellMask deep red (1:250, 10 µL, ThermoFisher Scientific, Pittsburgh, PA, USA, C10046) for 15 min at room temperature; a dye control was made by diluting into DPBS. To remove excess dye, ASC-EVs and dye control were subjected to UC at 100,000× *g* for 2 h (Microultracentrifuge: Sorvall Discovery, M150 SE, Rotor: S100-AT4, Pittsburgh, PA, USA). EVs were resuspended in cold DPBS (1250 µL). Potassium phosphate monobasic and dibasic were added to the DPBS to create a 2M buffer solution (pH = 8.0) for SIPE fabrication. SIPEs were fabricated as described in Section 2.4.

For SIPEs (1 × 10^9^ particles/mL, TFF isolated) co-cultured with SMCs, 100 µL was loaded into 0.4 µm transwell inserts (Greiner, Monroe, NC, USA, 662641) for co-culture with SMCs in fibrin gels and were cultured for either 6 or 24 h to allow for release from SIPEs and subsequent cellular uptake. Fluorescent expression as a Cy5 signal was interpreted as ASC-EV release and subsequent uptake. Following incubation, cells were fixed and stained according to previous protocols [52]. Imaging was conducted with an EVOS M7000 fluorescent imaging system (ThermoFisher Scientific, Pittsburgh, PA, USA,). Additional details can be found in the Appendix A.

### 2.9. Real-Time Quantitative Polymerase Chain Reaction (RT-qPCR) of Smooth Muscle Cells in Fibrin Gels

SMCs were prepared and resuspended in fibrin gels as previously stated after 3 days of culture [20,51]. The ASC-EVs at a concentration of 3 × 10^9^ particles/mL used in this assay for treatment and incorporation into SIPEs were purchased from RoosterBio Inc. (Frederick, MD, USA). SIPEs (using the same concentration of ASC-EVs) and SIMPs were co-cultured with SMCs using 0.4 µm transwell inserts. RNA was harvested from constructs using 1mL of TRIzol reagent (ThermoFisher Scientific, Pittsburgh, PA, USA, 15596026) before homogenization by pipetting up and down multiple times, until no visible pieces of gel remained. RNA was extracted using a PureLink™ RNA mini kit (ThermoFisher Scientific, Pittsburgh, PA, USA, 12183018A) following the manufacturer’s instructions. Isolated RNA was quantified using a NanoDrop™ OneC Microvolume UV-Vis spectrophotometer (ThermoFisher Scientific, Pittsburgh, PA, USA, 13-400-519) before conversion to cDNA using a high-capacity RNA-to-cDNA™ kit (ThermoFisher Scientific, Pittsburgh, PA, USA, 4387406) following the manufacturer’s instructions. RT-qPCR was performed using a QuantStudio 3 real-time PCR system (ThermoFisher Scientific, Pittsburgh, PA, USA, A26336) using a master mix comprising final concentrations of 1X PowerTrack™ SYBR Green Master Mix (ThermoFisher Scientific, Pittsburgh, PA, USA, A46109), 0.5 µM forward/reverse primer solution, and dH_2_O. A total of 10 ng of cDNA was used per reaction. Each experiment was performed in triplicate using n = 3 replicates. The primer sequences used are outlined in **Table 1**.

### 2.10. Fibrin Gel System for Vascular Smooth Muscle Cells for Long-Term Matrix Deposition

Fibrin gels were made as previously described [20,51]. Cells were left overnight in SBM to acclimate. The next day, the SBM was removed and replaced with treatment media. The treatment media consisted of 1 mL of DMEM/F-12 (Gibco, Gaithersburg, MD, USA, 12500062) with 10% FBS (Atlanta Biologics, Flowery Branch, GA, USA, S11550), 1% pen/strep (Gibco, Gaithersburg, MD, USA, 15070063), 12 mM ε-aminocaproic acid (Fluka, Charlotte, NC, USA, 07260), and 50 µg/mL of ascorbic acid (Millipore Sigma, St. Louis, MO, USA, A5960). For treatment with purchased ASC-EVs (1 × 10^9^ particles/mL, RoosterBio Inc., Frederick, MD, USA), SIPEs (fabricated with 1E9 particles/mL), and SIMPs, 100 µL of SIPEs or SIMPs was added to transwell inserts, or 10 µL of free ASC-EVs was added. A no-treatment control group (NT) was also used, where nothing was added to the media. The media was changed every 2–3 days, with new ASC-EVs added each media change. SMCs were cultured for 28 days, removed, and frozen for ninhydrin and hydroxyproline assays.

### 2.11. Ninhydrin (Insoluble Elastin) and Hydroxyproline (Collagen) Assays

Immediately following removal from cork-bored plates, fibrin gel constructs were frozen at −80 °C without fixation. According to previous protocols [20,51,53,54], tissue samples were thawed immediately before base hydrolysis (0.1 M NaOH, 1 h, 98 °C) followed by subsequent centrifugation (3000× *g*) to separate insoluble elastin protein from soluble non-elastin proteins. Acid hydrolysis (6 N HCl, 24 h, 110 °C) and assay quantification were then performed on both soluble and insoluble fractions (ninhydrin-based for elastin and hydroxyproline-based for collagen) to quantitatively determine ECM content in the fibrin gel constructs (reported as % elastin or collagen relative to total protein).

### 2.12. Development of Fibrin Gel Co-Culture System for Vascular Smooth Muscle Cells and U937 Cells

Fibrin gels were made as previously described using SMCs [20,51,53]. Gels were allowed to polymerize for 10 min at room temperature and 45 min at 37 °C before adding 1 mL of DMEM (Gibco, Gaithersburg, MD, USA, 12100-038) with 1% FBS (Atlanta Biologics, Flowery Branch, GA, USA, S11550), 1% pen/strep (Gibco, Gaithersburg, MD, USA,), and 12 mM ε-aminocaproic acid (Fluka, Charlotte, NC, USA). An adapted protocol from Airhart et al. was used for U937 cell culture [55]. Approximately 5 million U937 cells were plated into a T75 flask with 10 mL of supplemented RPMI media. The next day, U937 cells were activated with phorbol 12-myristate-13-acetate (PMA) (10 nM) for 24 h. Activated cells were passaged and resuspended in the low-serum DMEM for culturing SMCs. Approximately 24,000 cells/well of U937 cells were added to the fibrin gel constructs and left overnight to settle before adding treatments.

U937 cell and SMC co-culture media were changed to treatment media (same low-serum media as described previously with lipopolysaccharides (20 µg/mL, eBioScience, San Diego, CA, USA, 00-4976-03), and 100 µL treatments were added either directly to the media (purchased ASC-EVs at 1 × 10^9^ particles/mL from RoosterBio Inc., Frederick, MD, USA,) or in transwell inserts (SIPEs fabricated with 1 ×10^9^ particles/mL and SIMPs). The media was changed on day 3, 5, and 7 and replenished with fresh EVs each time. Removed media was centrifuged at 2500× *g* for 5 min to remove dead cells, and stored at −80 °C.

### 2.13. Zymogram to Monitor Matrix Metalloproteinase (MMP) Activity

Acrylamide gels (7.5%) with porcine skin gelatin (1.06 mg/mL, Millipore Sigma, St. Louis, MO, USA, G2500) were cast to create zymogram gels. Conditioned media collected from each media change as described in Section 2.12 (n = 3 for SIPE CM and SIMP CM, 15 µL) was mixed with Lamelli buffer (5 µL, Bio-rad, Hercules, CA, USA) and incubated at room temperature for 15 min. Non-conditioned media (nCM, media used for treatment of SMCs and U937 cells) was also used as a negative control for degradative activity. Samples (20 µL) were loaded into wells of the zymogram gel and run for 60 min at constant current (25 mA/gel). The gels were removed and washed three times in ultrapure water, two times in a 2.5% Triton-X solution for 15 min each, and another three times in ultrapure water. Gels were then incubated overnight (about 18 h) in development buffer (50 mM Tris-HCl (pH = 8.0), 5 mM CaCl_2_, and 1 µM ZnCl_2_) to activate the MMPs.

Following overnight incubation, gels were stained with 25 mL of Coomassie solution (0.1% *w*/*v* Coomassie brilliant blue, 10% *v*/*v* ethanol, 7.5% *v*/*v* acetic acid), rinsed with ultrapure water, and then destained overnight. Images were captured with white bands indicating MMP activity. ImageJ Version 1.54j was used to quantify the band intensity of each group.

### 2.14. SMC Migration Assay

SMCs were used to assess the biological effects of SIPEs on cell migration and proliferation. For migration assays, media consisting of 350 μL of experimental human SMC growth medium (10% supplement to 90% basal media, Cell Applications Inc, 311K-500) and 150 μL of treatment were added to the bottom of wells. Treatments included DPBS (treatment control), 3 × 10^9^ particles/mL ASC-EVs (same concentration used for loading into SIPEs, RoosterBio Inc., Frederick, MD, USA, E42059AD), SIMPs, SIPEs, supplemented SMC growth media (positive control), and experimental media (negative control). SMCs were resuspended in experimental media before plating 20,000 cells/0.3 mL in the top of transwell inserts (8 μm, Greiner, Monroe, NC, 662638). After 20 h of incubation at 37 °C and 5% CO_2_, transwell inserts were washed twice with PBS and fixed with 4% paraformaldehyde for 10 min. After washing with PBS, a tapered mini cotton swab (Medline Industries, Northfield, IL, USA, HDW826WCCZ) was used to remove non-migrated cells from the top of the transwell insert. Crystal violet dye was then added to each well for 5 min before being washed with ultrapure water and imaged on an EVOS M7000 fluorescent imaging system (ThermoFisher Scientific, Pittsburgh, PA, USA). The center of each transwell insert was captured by taking five 10× images, and ImageJ Version 1.54p was used to manually count migrated cells. Fold increase is presented as the average cells migrated per treatment relative to the average cells migrated with DPBS treatment (5 images per well with 3 wells per treatment for a total of 15 images per group).

### 2.15. SMC Proliferation Assay

For proliferation assays, SMCs were plated at 10,000 cells/0.5 mL using experimental media in 24-well plates. After letting cells attach overnight, the media was aspirated and 0.5 mL AlamarBlue (AB; ThermoFisher Scientific, Pittsburgh, PA, USA, DAL1025) at a 1:10 dilution in experimental media was added for a 1 h incubation. AB was then removed, and the excitation/emission (525/525 nm, respectively) was read on a fluorescent plate reader (Synergy HT, BioTek, Vermont, MA, USA) in duplicate (3 wells per treatment and 2 readings per well). Cells were washed twice with PBS before adding treatments. For treatments, 0.5 mL of experimental media was added into treatment wells before adding 0.4 µm pore transparent transwell inserts (Greiner, Monroe, NC, USA, 662641) with 150 μL of treatment. Treatments included DPBS, 3 × 10^9^ particles/mL, ASC-EVs (same concentration used for loading into SIPEs, E42059AD, RoosterBio Inc., Frederick, MD, USA,), SIMPs, and SIPEs, and fully supplemented media (positive control). AB readings were then taken at 48 h after removing the transwells and washing wells once with PBS. AB readings were also taken at 96 h after retreatment. All AB readings were normalized to blank wells, and fold increase was presented as fluorescent intensity per treatment relative to the fluorescent intensity of DPBS treatment.

### 2.16. Rat Model of AAA

All rats for these experiments were SAS Sprague Dawley strain, 240–260 g body weight, 8-week male rats (Charles River Laboratories, Garfield Heights, OH, USA). Animals were housed in a controlled animal facility, and all rat care and treatment occurred under protocols approved by the Vanderbilt University School of Medicine Institutional Animal Care and Use Committee (Protocol ID#: M2100022-00) and according to the ARRIVE guidelines.

Rat abdominal aortas were isolated and exposed according to previous protocols [56,57] through a ventral abdominal wall laparotomy. For rats receiving SIMP delivery, a subcutaneous microport (Instech, Plymouth Meeting, PA, USA) was connected to a polyurethane catheter tubing (Braintree Scientific, Inc, Braintree, MA, USA) and placed into the retroperitoneal space [4]. Type I porcine pancreatic elastase in saline (12 U/mL) (Millipore Sigma, St. Louis, MO, USA) was perfused for 30 min via iliac arteriotomy to place a polyethylene catheter for the perfusion. The elastase was perfused under constant pressure to the isolated infrarenal aorta using a syringe pump. After perfusion, an Ivalon sponge (5 × 8 mm) was connected to the tubing and placed over the top of the aorta. The incision was then closed, and the rats were allowed to recover. All aortas from all groups were harvested 13 to 15 d following elastase administration. The animal was sacrificed and the aorta from the proximal descending aorta to the iliac bifurcation was harvested. Only one rat was used per group to see initial proof of concept.

### 2.17. Preparation and Injection of SIMPs

SIMPs were mixed with bovine fibrinogen type 1 (200 µL, 33.3 mg/mL, Millipore Sigma, St. Louis, MO, USA, #8630,) and placed in a 1 mL syringe for shipment. Syringes were sealed and shipped at 4 °C to Vanderbilt University Medical Center to be used within 7 d of fabrication following washing. At 3 days post-surgery, the rat was placed on a custom magnet pad containing neodymium magnets (**Appendix A**) (S2, surface field ~0.88T, K&J Magnetics Inc, Pipersville, PA, USA). Then 200 µL of SIMPs in fibrinogen was mixed with 100 µL of thrombin (10 U/mL, Millipore Sigma, St. Louis, MO, USA, 605157-1KU) simultaneously through a custom 3D-printed bifurcation (**Appendix A**) and injected into the port of the rat (final concentration of fibrinogen: 22.2 mg/mL, final concentration of thrombin: 3.3 U/mL). The rat was left undisturbed for ~5 min so the hydrogel could polymerize.

### 2.18. Aorta Explant Staining

Explants were fixed in formaldehyde and washed 3 times with PBS (Gibco Gaithersburg, MD, USA). Aortas were then cut and placed in optimal cutting temperature medium (Fisher Healthcare, Pittsburgh, PA, USA, 4585) within cassettes. An electronic cryotome was used to slice 10mm sections onto superfrost plus microscope slides (ThermoFisher Scientific, Pittsburgh, PA, USA, 22034979). Slides were then heated on a slide warmer for 30 min before staining. Prussian blue stain was made by mixing equal parts of 7.4% HCl (J.T. Baker, Radnor, PA, USA, 9535-33) and 10% potassium ferrocyanide (Lab Chem, Zelienople, PA, USA, R5463200-500A). Slides were stained with 1 mL of stain for 30 min at room temperature. Slides were then washed 2 times with ultrapure water before adding 0.5 mL of nuclear fast red (Millipore Sigma, St. Louis, MO, USA, N3020) counterstain for 5 min. Slides were washed 3 times with ultrapure water and mounted using Permount mounting medium (Fisher Chemical, Pittsburgh, PA, USA, SP15-100) before imaging at 4×, 10×, and 20× on an EVOS cell imaging system.

### 2.19. Statistical Analysis

Statistical analyses were performed to determine significance between groups, with *p* < 0.05 used to represent significance. Paired *t*-tests were used to analyze differences in protein content between magnetic and non-magnetic fractions. An ANOVA with a post hoc Tukey’s test was used to determine differences between the concentration of protein in magnetic and non-magnetic fractions across the different microparticle groups. A matched two-way ANOVA between SIPEs and SIMPs over 7 days, with Šídák’s multiple comparisons test was used for flow cytometry analysis. For PCR, either ordinary one-way ANOVA with Dunnett’s multiple comparisons test (normal distribution/parametric test), or Kruskal–Wallis with Dunn’s multiple comparisons test (non-normal distribution/non-parametric test) was used. For matrix deposition, a Kruskal–Wallis with Dunn’s multiple comparisons test was used. For degradation, an unpaired *t*-test was used. For migration, ordinary one-way ANOVA with Tukey’s multiple comparison test (normal distribution/parametric test) was used. For proliferation, ordinary 2-way ANOVA with Tukey’s multiple comparison test was used. Results are reported as mean ± standard deviation (S.D.). All results were analyzed in GraphPad Prism 10 (GraphPad, Boston, MA, USA).

## 3. Results

### 3.1. EV and ASC Characterization

ASC-EVs purchased from RoosterBio Inc. (Frederick, MD, USA) were characterized. These ASC-EVs demonstrated an average diameter of 144.9 nm ± 47.4 nm (**Appendix A**); ASC-EVs 3 × 10^9^ particles/mL (26.00 ± 0.10 µg/mL) lysed with RIPA buffer had more protein compared to non-lysed samples at a concentration of 3 × 10^9^ particles/mL (9.05 ± 0.50 µg/mL) (**Appendix A**). ASC-EVs were negative for apolipoprotein B via Western blot (**Appendix A**) and positive for CD63, EpCAM, ANXA5, TSG101, GM130, FLOT1, ICAM, ALIX, and CD81 (Appendix A). Characterization provided by RoosterBio Inc (Frederick, MD, USA) showed results related to particle sizes (Appendix A), protein concentration per 1 × 10^9^ particles/mL (Appendix A), EV purity (Appendix A), and surface markers CD9 (Appendix A), CD63 (Appendix A), CD81 (Appendix A), ALIX (Appendix A), and TSG101 (Appendix A). RoosterBio Inc. (Frederick, MD, USA) also provided characterization of cytokines secreted by the ASCs used for ASC-EV isolation (**Appendix A**).

### 3.2. SEM of SIPEs

Following fabrication with the salting out method, SIPEs were able to be formed using EVs isolated via UC (**Figure 1**). Even in the presence of ASC-EVs, SIPEs demonstrated similar morphology and size to SIMPs.

### 3.3. Magnetic Separation Assay of SIPEs, SIMPs, Silk EV MPs, and Silk MPs

The amount of silk protein in the magnetic fraction of SIPEs trended towards higher protein content than that in the corresponding non-magnetic fraction (n = 3, 15.56 ± 7.22 µg vs. 0.00 ± 0.00 µg, *p* = 0.065) (**Figure 2**). Conversely, there was significantly less silk protein in the magnetic fraction of silk EV MPs compared to the non-magnetic fraction (n = 3, 0.00 ± 0.00 µg vs. 14.12 ± 3.08 µg, *p* = 0.0155). These results indicate that SIPEs are magnetically moveable.

### 3.4. Flow Cytometry of SIPE Releasates

Comparing the percentage of CD63+ events between SIPEs and unloaded SIMPs demonstrated relatively higher numbers in SIPE releasates at day 1. Over 7 days, only SIPEs showed release of CD63+ events and only within the first day (**Figure 3**).

### 3.5. SIPE Released EV Uptake at 6 H vs. 24 H

Following either 6 h or 24 h of incubation with SMCs in fibrin gels, SIPEs were able to release fluorescently labelled EVs, as evidenced by cellular uptake. SIPE-released ASC-EVs had more fluorescent signal in the Cy5 channel at 24 h compared to that at 6 h (**Figure 4A,B**). SIMPs (**Figure 4C,D**), which were loaded with a dye control, had little to no Cy5 signal at both 6 and 24 h. Purchased ASC-EVs were also able to be released and taken up by SMCs (**Appendix A**).

### 3.6. Real-Time Quantitative PCR of SMC Seeded Fibrin Gels

After 3 days of culture with DPBS, SIMPs, SIPEs, ASC-EVs, or fully supplemented media, there were minor differences in the gene expression of certain elastogenesis-related genes (ELN, FBN1, LOX, EFEMP2, and FBLN5, *p* > 0.05) (**Figure 5**). Col1A1 was significantly increased in gels treated with ASC-EVs (2.093 ± 0.467-fold change, *p* = 0.0120), but SIPEs did not have the same effect (1.150 ± 0.195-fold change, *p* = 0.9672). LOXL1 was also significantly increased in groups treated with SIMPs (2.147 ± 0.237-fold change, *p* = 0.0007), ASC-EVs (1.903 ± 0.343-fold change, *p* = 0.0043), and fully supplemented media (2.013 ± 0.180-fold change, *p* = 0.0019) and trended toward an increase when treated with SIPEs (1.543 ± 0.199-fold change, *p* = 0.0763).

### 3.7. SIPE and ASC-EV Effects on Matrix Deposition

Neither ASC-EVs encapsulated in SIPEs (ELN: 0.90 ± 0.60%, COL: 0.31 ± 0.37%) nor free ASC-EVs (ELN: 0.61 ± 0.28, COL: 0.21 ± 0.31%) stimulated more elastin or collagen synthesis compared to SIMPs (ELN: 1.43 ± 1.62%, COL: 0.44 ± 0.63%) and the NT (ELN: 0.75 ± 0.49%, COL: 0.44 ± 0.65%) treatment group (**Figure 6**).

### 3.8. SIPE and EV Effect on Degradative Activity

Zymography of the CM did not demonstrate any significant differences in Pro-MMP9 and MMP2 activity between the SIPE CM and SIMP CM (**Figure 7**) (Pro-MMP9: 5612 ± 608 vs. 4709 ± 1028 band intensity units, *p* = 0.2608 and MMP2: 6542 ± 631 vs. 5113 ± 996 band intensity units, *p* = 0.1038). Conversely, there was a significant increase in Pro-MMP2 activity in SIPE CM compared to SIMP CM (92935 ± 4300 vs. 72334 ± 11104 band intensity units, *p* = 0.0401).

### 3.9. Migration and Proliferation of SMCs

SMCs demonstrated significantly more migration towards ASC-EV treatment compared to DPBS treatment (77.31 ± 19.30 times vs. 1.00 ± 0.85, *p* ≤ 0.0001) and SIPE treatment (1.12 ± 0.44 times, *p* ≤ 0.0001) (**Figure 8A**). There was no significant difference between SIPE treatment compared to DPBS treatment and SIMP treatment (both *p* > 0.9999).

SMCs proliferated 1.22 ± 0.01 and 1.20 ± 0.09 times above DPBS-treated cells when treated with ASC-EVs and SIPEs, respectively, after 48 h (*p* ≤ 0.01 and 0.05, respectively) (**Figure 8B**). After 96 h, ASC-EVs and SIPEs increased SMC proliferation by 1.29 ± 0.11 and 1.12 ± 0.05 times, respectively, compared to DPBS-treated cells (*p* ≤ 0.001 and *p* > 0.05, respectively). SIMPs had decreased proliferation after 96 h compared to DPBS-treated cells (0.97 times). All fold changes for each assay can be found in **Table 2**, and figures with all groups can be found in **Appendix A**.

### 3.10. Magnetic Localization of SIMPs in an In Vivo AAA Rat Model

AAA samples (n = 1) treated with SIMPs demonstrated localization of the particles around the aorta, indicated by the Prussian blue stain; saline-treated aortas not treated with SIMPs showed less blue coloration (**Figure 9**).

## 4. Discussion

Encapsulation of ASC-EVs into SIPEs did not disrupt their magnetic moveability (**Figure 2**), which was expected as the EVs did not disrupt fabrication of the particles (**Figure 1**), nor did they change their size or morphology. CD63+ events (reflecting the presence of an established EV surface protein marker) were able to be detected with flow cytometry and demonstrated burst release within 1 day (**Figure 3**), indicating that the SIPEs were able to release ASC-EVs. As expected, released EVs had a more vibrant signal of Cy5 compared to dye control when taken up by SMCs (**Figure 4**). Since the ASC-EVs had to first be released by the SIPEs, it was expected that it would take longer for released ASC-EVs to be taken up than free ASC-EVs; others have shown uptake within 6 h of culture [52]. Neither ASC-EVs nor SIPEs had strong effects on matrix-related genes (**Figure 5**), indicating that these ASC-EVs were not elastogenic in general. This conclusion was further demonstrated using protein quantification, in which neither SIPEs nor ASC-EVs increased insoluble elastin or collagen content (**Figure 6**). Thus, it is reasonable to assume that SIPEs containing these ASC-EVs would not have an elastogenic effect either. Additionally, there was no decrease in degradative activity (**Figure 7**), indicating that these ASC-EVs and SIPEs were not effective in anti-degradative activity. SMC migration assays showed a clear deficit in the SIPEs compared to the free EVs (**Figure 8**). As the concentrations of ASC-EVs loaded into SIPEs and free ASC-EVs were the same, the loading efficiency of the EVs may be insufficient to solicit a migratory response. Proliferation-wise, the SIPE-released EVs had a similar effect to that of free ASC-EVs within 48 h. Since the flow cytometry showed that most of the ASC-EVs were released within 24 h, it can be assumed that the most potent effects of the ASC-EVs were within that 24 h window. Finally, SIMPs were able to be localized within a rat model of AAA (**Figure 9**), demonstrating proof of concept for eventual delivery.

Given that the fabrication method used for making SIPEs was adapted from a previous study [37], it was reasonable that the release kinetics might be similar. Kucharczyk et al. [37] demonstrated slow release of doxorubicin over a period of 7 days; however, we saw a burst release of ASC-EVs within the first 24 h. As EVs are much larger than a small drug molecule like doxorubicin, loading within the particles could be more difficult and have lower loading efficiency. Their particles were also much smaller than ours, with an average size of 0.5 µm compared to ours, which were approximately 1 µm in size based on our SEM images (**Figure 1**). Additionally, they tested the effects of different pHs on release, which could potentially affect the release of various cargos, including small drugs and larger cargos like ASC-EVs. They also used spectrophotometry for release analysis, which required less downstream processing and was less sensitive than the flow cytometry used in our study. Bari et al. [58] loaded EVs into silk fibroin-based scaffolds mixed with sodium alginate and found that silk fibroin induced slower release kinetics than our silk-iron particles, indicating that potential biomaterial hybrids could be investigated to fine-tune release.

For free EVs, uptake can occur in as little as 1 h in SMCs [59], depending on the parent cell of the EVs. Sajeesh et al. [27] reported MSC-EV uptake in SMCs with EVs after 2 h of culture while Niu et al. [59] saw uptake in 1 h with macrophage-derived EVs. Thus, it was reasonable that EVs encapsulated in SIPEs needed more time to release the EVs for eventual uptake. In a previous study from our lab [20], we tested free EVs on their ability to stimulate elastogenesis. This study only showed increased expression of fibrillin-1 when treated with EVs, which was not seen in our current study. However, others [28] who have treated SMCs with bone marrow-derived mesenchymal EVs saw increased expression of ELN, FBLN5, LOX, and LOXL1. Thus, the source of stem cells may play a role in their elastogenic effects.

EVs were isolated from a single cell donor, which did not account for potential interdonor variability. While there were different donors for the purchased ASC-EVs and isolated ASC-EVs, these two different groups were not tested. Additionally, since the cells were from a single donor, sex-based differences on ASCs could also not be assessed [60], which could potentially affect the EVs. There were two different isolation techniques used (UC and TFF) along with commercially available EVs, which could potentially give different subpopulations of EVs [61] and different purity levels. Additionally, EVs contain heterogenous subpopulations [62], which could be further exacerbated by different isolation methods and donors. Moving forward, our group intends to use only rigorously characterized EVs from RoosterBio Inc. (Frederick, MD, USA) (which internally performs TFF isolations) to standardize the EV isolation process.

The magnetic separation was only performed on one batch of SIPEs using ninhydrin analysis. It still needs to be determined whether EVs were retained within the SIPEs; this could potentially be performed with a lipid assay analysis, fluorescent EV detection, or flow cytometry to detect for CD63 or some other positive EV marker. Though the uptake assay performed with both in-house EVs and purchased RoosterBio Inc. EVs (Frederick, MD, USA) clearly indicated the encapsulation and subsequent release of EVs, a quantitative metric of EV release should also be developed to further optimize EV release, which could potentially be achieved with flow cytometry of the EVs themselves not attached to magnetic beads. This quantitative metric could also be used to assess the loading efficiency of EVs; by performing flow cytometry of fluorescently stained EVs positively stained for CD63 or other EV markers, EVs could be quantified before encapsulation. The EVs could then be released through degradation and quantified again to see how many EVs were encapsulated. EV potency was only validated with one batch of EVs and SIPEs and should be measured further to see their effects across different batches. Our future studies intend to incorporate these changes for further characterization of the release profile and encapsulation efficiency. In terms of any paracrine effects of the EVs, our study does not identify specific cytokines contained within the EVs nor does it compare them to the parent cells, which have been quantified by RoosterBio Inc. (Frederick, MD, USA, **Appendix A**). Others [24,63,64] have identified different cytokines and RNA within EVs to identify how EVs affect different target cells. Our future experiments should include EV cargo analysis and mechanistic approaches to better understand the full effect of EVs on SMC activity. Additionally, the in vivo aspect was only used with one rat and with SIMPs for proof of concept. Thus, more rats should be used to further validate the delivery system and potentially quantify the amount of particles that are localized around the aorta.

As the EVs were able to be taken up by SMCs in 3D fibrin gel constructs, future studies should investigate whether EVs can penetrate thicker tissues (such as the aorta itself). As our study was successful in demonstrating that 3D fibrin constructs were not a barrier to EV uptake, another logical next step would be to test EV uptake in vivo as well. Our eventual goal is to deploy SIPEs within a hydrogel, specifically fibrin, locally and peri-adventitially to the site of an AAA. Thus, fluorescently stained SIPEs should be injected within a fibrin gel around an AAA in a rat and imaged to see if EVs are able to penetrate the aortic wall. If the EVs cannot penetrate the wall to be taken up by SMCs, it is still worth attempting to see where the EVs are taken up. If EVs are taken up by cells within the adventitial layer, then more tests should be performed to establish potential paracrine effects between adventitial and medial cells.

Though the EVs were taken up by SMCs, the specific mechanism of this EV uptake is yet to be studied. As described above, if the therapeutic EVs are unable to reach the SMCs in vivo but still have a regenerative effect, then it is important to see the mechanisms by which EVs could be taken up by SMCs and other vascular cells. Others [65] have demonstrated cross-talk via EVs between SMCs and other vascular cells. Therefore, if there is a paracrine effect, then the EVs could be taken up by adventitial cells and trigger release of other EVs, cytokines, etc., that can affect SMCs. There are a variety of EV uptake mechanisms described, including clathrin-mediated endocytosis, caveolin-dependent endocytosis, micropinocytosis, and phagocytosis [66,67]. By elucidating the mechanism by which these ASC-EVs are taken up by SMCs and other resident cells, a better understanding for treatment of aneurysm could be established.

## 5. Conclusions

SIPEs were shown to be magnetically moved and separated into magnetic and non-magnetic fractions. SIPEs were also shown to release EVs, which could consequently be taken up by SMCs. While SMCs were able to take up the released EVs, the EVs did not have a potent enough effect to stimulate similar migration numbers to free EVs. Proliferation-wise, the SMCs were equally affected by SIPEs and free EVs within 48 h. Neither SIPEs nor free EVs demonstrated significant increases in elastogenic gene expression. Finally, SIMPs could be localized and imaged around the aorta in an in vivo model of AAA.

## 6. Patents

The technology used in this research was filed as a provisional patent in the U.S., titled “Silk Particles for use in Tissue Engineering, and Related Methods,” Serial No.: 63/535,695.

## Figures and Tables

**Figure 1 jfb-16-00395-f001:**
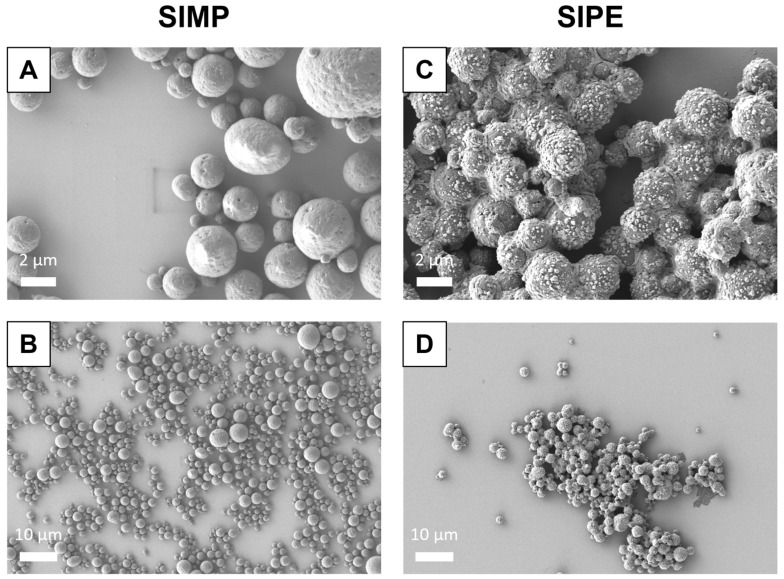
**SEM image of SIMPs (A and B) and SIPEs (C and D) at both high (5000×) and low magnification (1000×)**. Both SIMPs and SIPEs were fabricated using a salting out method using a potassium phosphate buffer solution, with ASC-EVs mixed into the potassium phosphate buffer to fabricate SIPEs. They were then imaged using scanning electron microscopy (SEM). SIMPs and SIPEs both displayed round morphology and similar size when using this fabrication method.

**Figure 2 jfb-16-00395-f002:**
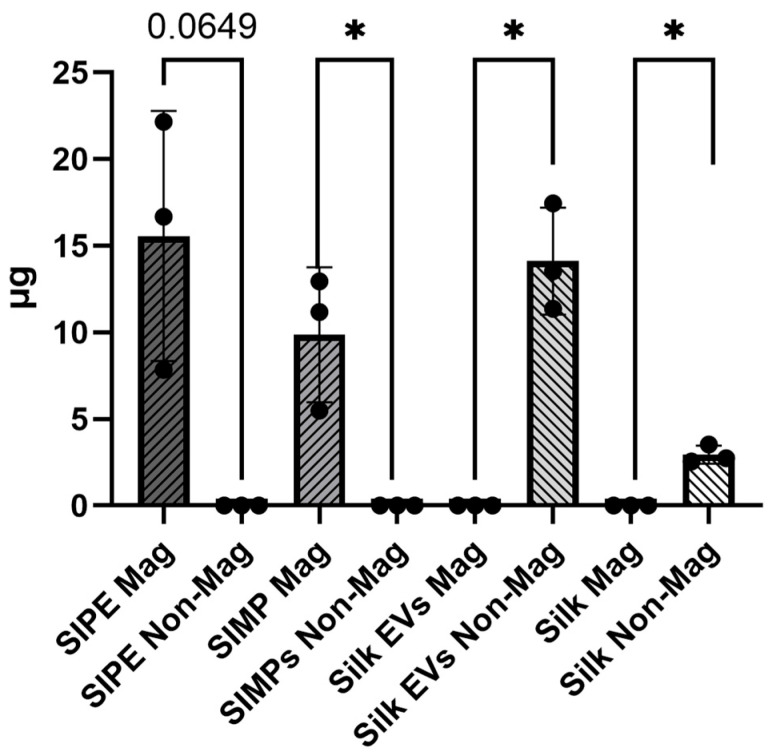
**Both SIPEs and SIMPs have more protein in their magnetic fraction compared to their non-IONP containing counterparts (silk EVs and silk), indicating magnetic moveability of these MPs.** There was more silk protein in the magnetic fraction of the SIPEs compared to that in the non-magnetic fraction (n = 3, 15.56 ± 7.22 µg vs. 0.00 ± 0.00 µg, *p* = 0.065). The silk EV group had significantly less protein in its magnetic fraction compared to the non-magnetic fraction (n = 3, 0.00 ± 0.00 µg vs. 14.12 ± 3.08 µg, *p* = 0.0155). Similar results were also seen in the SIMP and silk MP groups (SIMPs: n = 3, magnetic fraction: 9.87 ± 3.89 µg vs. non-magnetic fraction: 0.00 ± 0.00 µg, *p* = 0.048; Silk MPs: n = −3, magnetic fraction: 0.00 ± 0.00 vs. non-magnetic fraction: 2.94 ± 0.52, *p* = 0.010). * = *p* ≤ 0.05.

**Figure 3 jfb-16-00395-f003:**
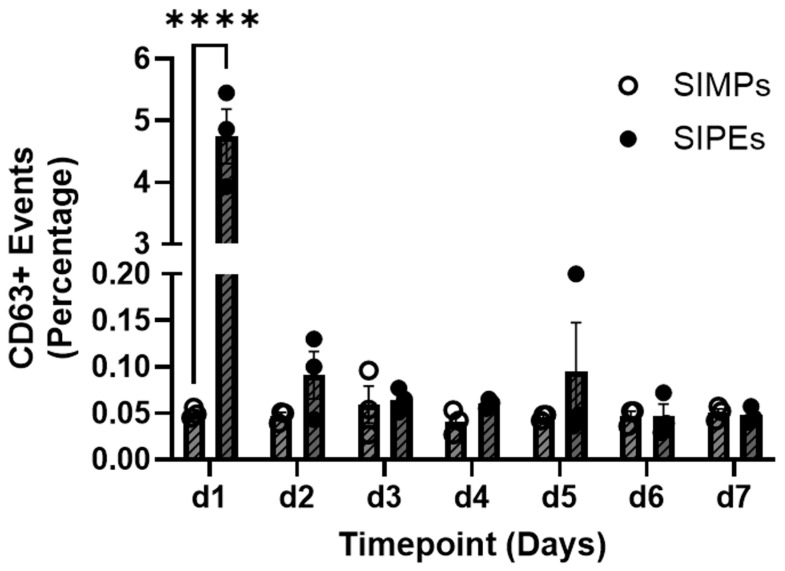
**Flow cytometry of CD63+ events demonstrated burst release of ASC-EVs from SIPEs within 1 day, but relatively no release afterward.** SIPEs showed the highest percentage of CD62+ events within one day (4.747 ± 0.766%, *p* < 0.0001) compared to the SIMPs and showed relatively negligible percentages of CD63+ events (0.050 ± 0.006%) within one day. After that first day, the percentage of CD63+ events released from SIPEs stayed below 1%. **** = *p* ≤ 0.0001.

**Figure 4 jfb-16-00395-f004:**
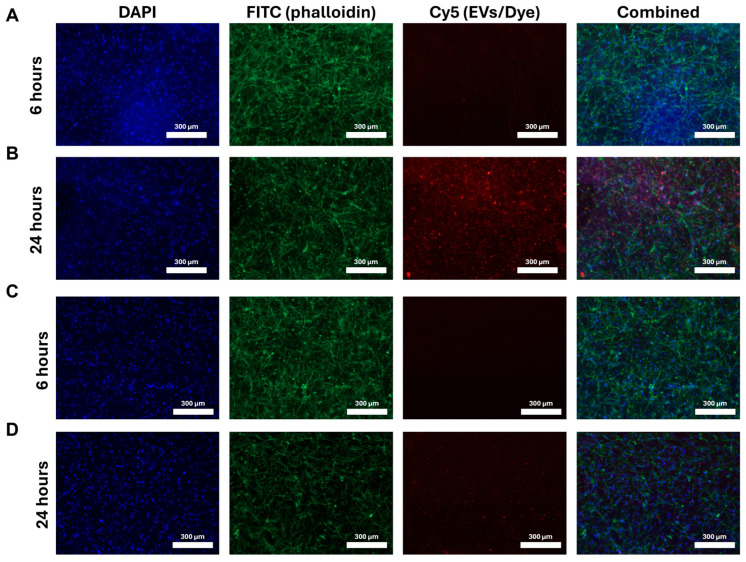
**SIPEs released more ASC-EVs in 24 h compared to 6 h and had higher signal in the Cy5 channel compared to SIMPs.** After co-culture with transwell inserts, SIPEs had more Cy5 signal (and therefore ASC-EVs) at 24 h (**B**) compared to that at 6 h (**A**). SIMPs had little to no signal in the Cy5 channel at both 6 h (**C**) and 24 h (**D**).

**Figure 5 jfb-16-00395-f005:**
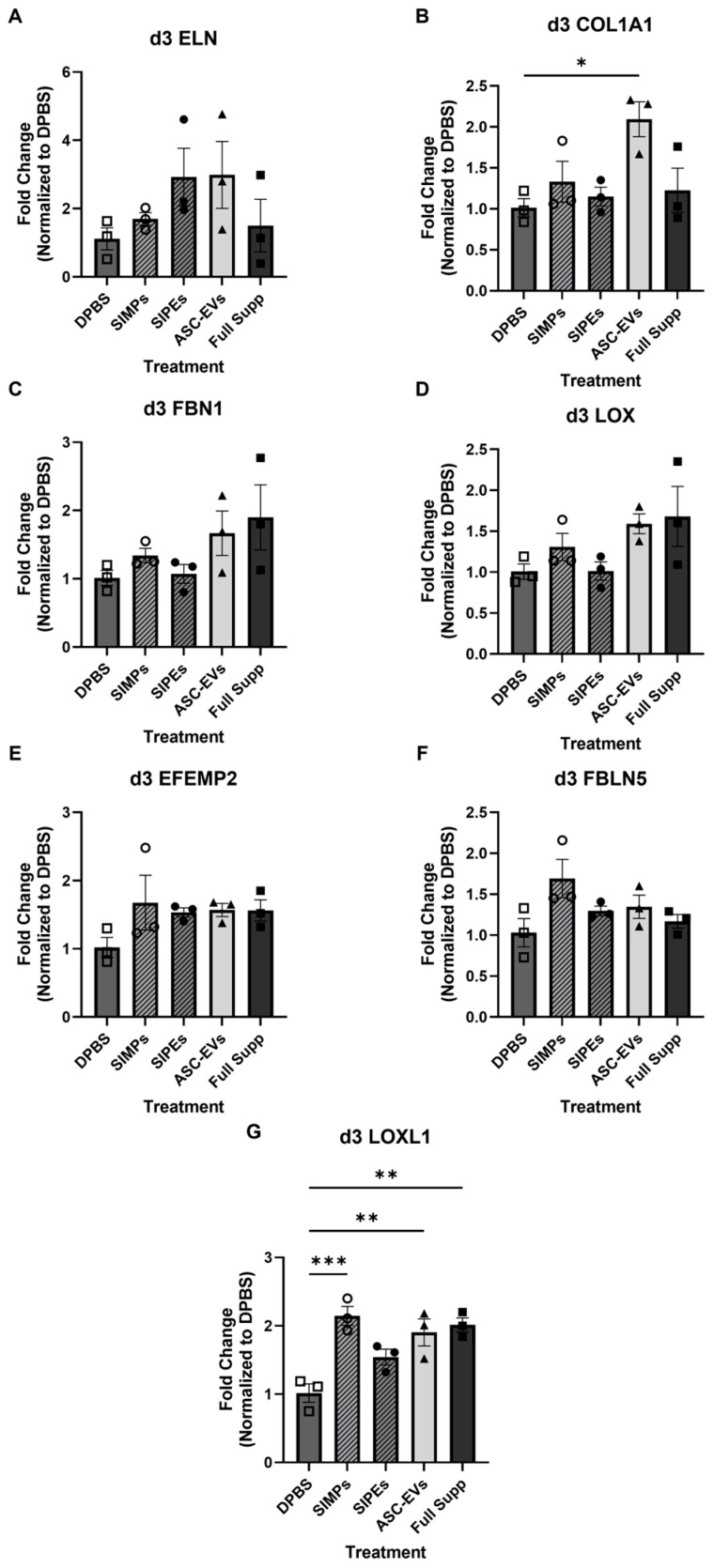
**SIPE-treated gels did not demonstrate increased gene expression of elastogenesis-related genes (tropoelastin (ELN, (A)), collagen 1 A1 (COL1A1, (B)), fibrillin-1 (FBN1, (C)), lysyl oxidase (LOX, (D)), EGF-containing fibulin extracellular matrix protein 2 (EFEMP2, (E)), fibulin-5 (FBLN5, (F)), and lysyl oxidase-like 1 (LOXL1, (G))) following 3 days of culture.** Fibrin gels were treated with DPBS, SIMPs, SIPEs, ASC-EVs, and fully supplemented media. Elastogenic genes including ELN, FBN1, LOX, l EFEMP2, and FBLN5 did not increase in expression when treated with ASC-EVs or SIPEs (*p* > 0.05). COL1A1 expression was increased with ASC-EV treatment (2.093 ± 0.368-fold change, *p* = 0.0120) but not with SIPEs treatment (1.150 ± 0.195-fold change, *p* = 0.9672). LOXL1 expression was increased with SIMPs (2.147 ± 0.237-fold change, *p* = 0.0007), ASC-EVs (1.903 ± 0.343-fold change, *p* = 0.0043), and fully supplemented media (2.013 ± 0.180-fold change, *p* = 0.0019) and trended toward an increase with SIPEs (1.543 ± 0.199-fold change, *p =* 0.0763). All data was analyzed with respect to fold change and normalized to DPBS; samples were performed in triplicate. * = *p* ≤ 0.05, ** = *p* ≤ 0.01, *** = *p* ≤ 0.001.

**Figure 6 jfb-16-00395-f006:**
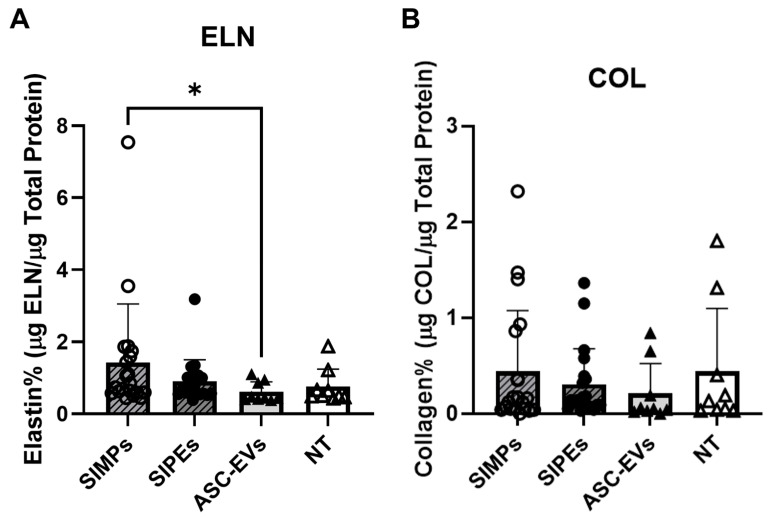
**Purchased ASC-EVs were used to fabricate SIPEs and as free ASC-EVs but showed no significant differences in elastin or collagen deposition.** (**A**) There were no significant differences in elastin deposition between SIPEs (0.90 ± 0.60%, n = 20), SIMPs (1.43 ± 1.62%, n = 20), ASC-EVs (0.61 ± 0.28, n = 9), and NT (0.75 ± 0.49%, n = 9) except between SIMPs and ASC-EVs (*p* = 0.0343). (**B**) There were no significant differences in collagen deposition (*p* > 0.05) between SIPEs (0.31 ± 0.37%), SIMPs (0.44 ± 0.63%), ASC-EVs (0.21 ± 0.31%), and NT (0.44 ± 0.65%). * = *p* ≤ 0.05.

**Figure 7 jfb-16-00395-f007:**
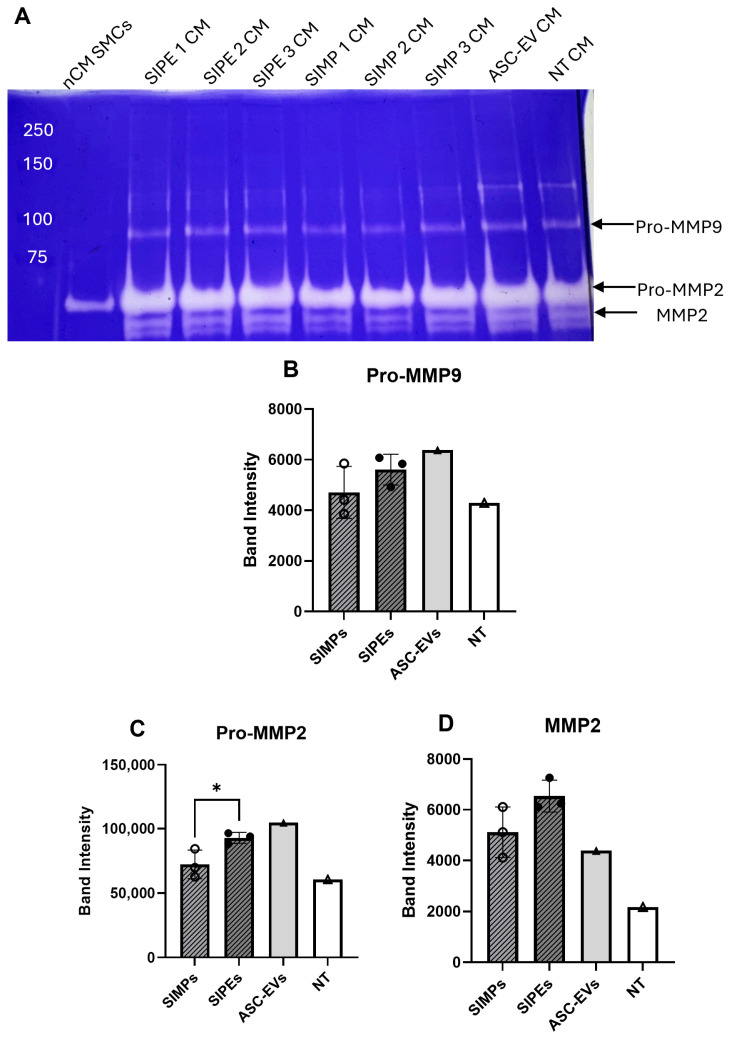
**Zymogram of CM collected from SMCs treated with SIMPs, SIPEs, ASC-EVs, or no treatment did not show decreased degradative activity.** (**A**) Zymogram of the different treatment groups (SIPE-treated CM, ASC-EV-treated CM, SIMP-treated CM, and no treatment (NT)) and non-conditioned media (nCM) used to treat SMCs showed some degradative activity in all groups. (**B**) Approximate band intensity of Pro-MMP9 for the four groups, with SIPE-treated CM having higher Pro-MMP9 band intensity compared to SIMP-treated CM (5612 ± 608 vs. 4709 ± 1028 band intensity units, *p* = 0.2608. (**C**) Approximate band intensity of Pro-MMP2 for the four groups, with SIPE-treated CM having higher Pro-MMP2 band intensity compared to SIMP-treated CM (92,935 ± 4300 vs. 72,334 ± 11,104 band intensity units, *p* = 0.0401). (**D**) Approximate band intensity of MMP2 for the four groups, with both SIPE-treated CMs having higher MMP2 band intensity compared to SIMP-treated CM (6542 ± 631 vs. 5113 ± 996 band intensity units, *p* = 0.1038). * = *p* ≤ 0.05.

**Figure 8 jfb-16-00395-f008:**
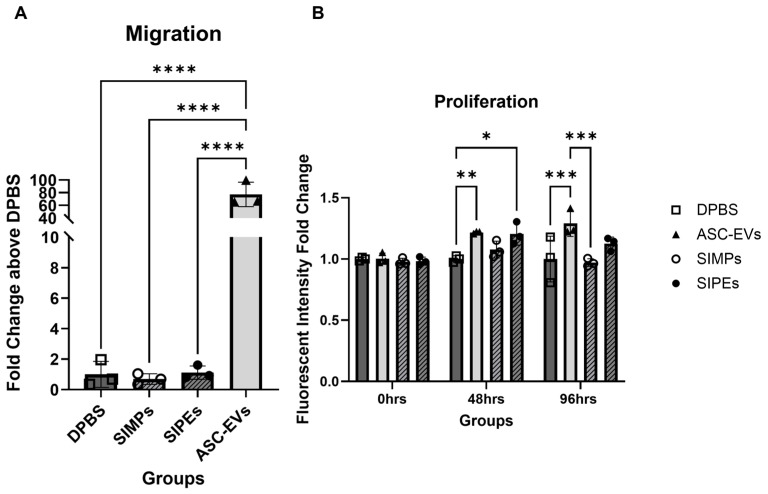
**Free ASC-EVs but not SIPEs increased SMC migration (A) and both increased SMC proliferation within 48 h (B).** When SMCs were treated with free EVs, they migrated significantly more compared to all other groups. EVs also increased proliferation of SMCs at both 48 and 96 h, while SIPEs only increased proliferation after 48 h.* = *p* ≤ 0.05, ** = *p* ≤ 0.01, *** = *p* ≤ 0.001, **** = *p* ≤ 0.0001.

**Figure 9 jfb-16-00395-f009:**
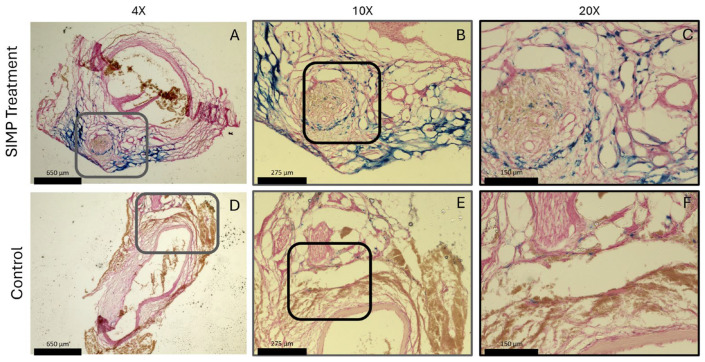
**AAA samples showed localization of SIMPs when exposed to a magnet.** SIMP-treated aortas displayed blue stained iron using Prussian blue staining surrounding the aorta (**A**–**C**) compared to saline-treated control samples that showed little to no blue stain (**D**–**F**). Gray boxes in (**A**,**D**) represent the zoomed in region in (**B**,**E**); black boxes in (**B**,**E**) represent the zoomed in region of (**C**,**F**). Both aortas displayed red nuclei with pink non-specific nuclear fast red stain to visualize the aortic tissue.

**Table 1 jfb-16-00395-t001:** **Primers used for RT-qPCR**.

Gene	Forward Primer	Reverse Primer
ELN (Tropoelastin)	AGTTGGCATTTCCCCCGAAG	TAACCCAAACTGGGCGGCTT
COL1A1	AGTGTGGCCCAGAAGAACTG	CCGCCATACTCGAACTGGAA
FBN1	CGTCAACACTGATGGCTCCT	CTCCGCATGTGTGTGTCAAC
LOX	CGACCCTTACAACCCCTACA	CAGGTCTGGGAGACCGTACT
EFEMP2	GCCCGAGTGTGTGGACATAG	ACACAGGAGCGGTTGTTAGG
FBLN5	TTCTTCTCGCCTTCGCATCT	ATTCGTGCACTGTGCCTGT
LOXL1	CAGACTTCCTCCCCAACCG	ATGCTGTGGTAATGCTGGT
RPS9	CTGAAGCTGATCGGCGAGTA	GGGTCCTTCTCATCAAGCGT

**Table 2 jfb-16-00395-t002:** **Fold change in SMC migration and proliferation at 48 and 96 h when treated with DPBs,** **ASC-EVs,** **SIMPs,** **and** **SIPEs.**

	DPBS	ASC-EVs	SIMPs	SIPEs
Migration	1.00 ± 0.85	77.31 ± 19.30	0.69 ± 0.35	1.12 ± 0.44
Proliferation 48 h	1.00 ± 0.03	1.22 ± 0.01	1.08 ± 0.07	1.20 ± 0.09
Proliferation 96 h	1.00 ± 0.19	1.29 ± 0.11	0.97 ± 0.03	1.12 ± 0.05

## Data Availability

The original contributions presented in the study are included in the article, further inquiries can be directed to the corresponding author.

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
