# Peer review of "“Attractive” Treatment for Abdominal Aortic Aneurysm Repair: Magnetic Localization of Silk-Iron Packaged Extracellular Vesicles"

_jfb, 2025, doi:10.3390/jfb16110395_

Round 1
Reviewer 1 Report
Comments and Suggestions for Authors
This MS proposed a therapeutic strategy for abdominal aortic aneurysm repair using silk-iron incorporated extracellular vesicles (SIPEs). The following comments need to be addressed or clarified to enhance the MS.
- Sections (2.3 and 2.4) describe the methodology of fabricating and the incorporation of vesicles. However, it lacks sufficient detail; readers must refer to multiple cited prior published reports to piece together the methodology. Authors should revise these sections to provide a clear description of fabrication and incorporation processes, including all procedure details such as timing and concentrations, and create a schematic presentation to illustrate these steps. In addition, please clarify how magnetic and release properties were evaluated and validated.
- Quantitative data on extracellular vesicles (ASC-EV) incorporation, loading efficiency and their release kinetics are required to support the proposed delivery.
- The variability in extravesicular content between sources or isolation methods is common, its recommended to use batches to limit this problem, and also discuss the validation and the impact of it on reproucability
- Discuss the impact of using silk fibroin matrix on extracellular vesicles (ASC-EV), stabilization effect or regulating release properties
- Please be consistent in terminology (e.g. microparticles and SIPEs)
- Check grammatical and minor typos thorough the text
- The source and grade of all used materials should be provided (e,.g. silk fibroin, iron oxide)
- Please include the institutional animal care and use committee approval number
Reviewer 2 Report
Comments and Suggestions for Authors
The study investigates to incorporate ASC-EVs into silk-iron microparticle and examine their biological activity. The study is novel and interesting however some informations in the methodology section (especially some concentrations) is missing. Authors should give detailed information about all treatment and chemical doses in method section. (2.4 Fabrication of SIPEs and SIMPs section: ASC-EVs amount, 2.7 Flow Cytometry Releasate Experiments: SIPEs/SIMPs amount, 2.8 EV Uptake: SMCs and fibrin gels amount, etc)
Reviewer 3 Report
Comments and Suggestions for Authors
In this manuscript ““Attractive” treatment for Abdominal Aortic Aneurysm Repair: Magnetic Localization of Silk-Iron Packaged Extracellular vesicles”, Marini et al. described the synthesis of silk-iron packaged extracellular vesicles, and tested its effect to a variety of aspects, including EV uptake, expression of genes of extracellular matrix, ECM deposition, MMPs release, cell migration and proliferation. The manuscript is well described especially in materials and methods that is useful for readers. Unfortunately the results do not show effective regeneration compared with just EVs, however, reviewer think the work is worth published. Reviewer has several points to be revised mainly in EVs from Roosterbio.
Minor comments
- Introduction
The authors should describe previous work more in this section. “Silk-iron” is not easy to understand. Previous work seems to use IOP as a core, then coat silk fibroin outside. The authors should describe what is the core, what is coating, and for what silk fibroin should be used in Introduction. And for extracellular vesicles, the authors should describe how EVs are packaged; side-by-side conjugation between EVs and silk-IOP or EVs are inside of large silk-iron nanoparticles or silk-iron nanoparticles are inside of large extracellular vesicles.
- In Figure S6, the authors checked EV markers, but the data is not clear except positive control. In addition, reviewer believe, GM130 is the Golgi marker and should not be in secreted in EV fraction. In addition, EpCAM and ICAM1 are plasma membrane proteins. Probably microvesicles/ectosome are collected together with endosome-derived exosomes positive for CD63, CD81 ALIX and TSG101. Reviewer do not demand to isolate EVs again to the authors, as this is out of points of this paper, but the authors should describe which protein was observed and probably which membrane fraction is collected as EVs based on the data more precisely.
- In Figure S11 and S13, there are two peaks of CD63 and four peaks for ALIX. What does green square mean in the figure? Please describe it.
- Figure S15, the authors describe the cytokines. Are cytokines associated to EVs? If so, cytokine effects will be observed in addition to regenerative RNAs in EVs. Please discuss whether there is cytokine effects with EVs or not.
- Regarding figure 1 and 2, the size distribution of SIMP and SIPE should be described at least in text or supplementary figure. In Figure 4, the authors only looked at EVs, but is SIMP or SIPE located outside of SMC cells or internalized by SMC cells? Please discuss the possibility of internalization of SIMP or SIPE.
Reviewer 4 Report
Comments and Suggestions for Authors
Comments and Suggestions for Authors
The manuscript describes the fabrication, characterization and proof-of-principle testing of silk-iron packaged extracellular vesicles (SIPEs). The authors demonstrate fabrication of SIPEs by salting-out and incorporation of IONPs without altering microparticle morphology, magnetic moveability and magnetic separation of SIPEs, an early “burst” release and subsequent uptake of the released, fluorescently labelled EVs by vascular smooth muscle cells. Overall, the study is timely, well-structured and technically competent. The approach (combining a biodegradable protein carrier, magnetic targeting and EV therapeutics) addresses an important translational problem — local retention of EVs to avoid rapid systemic clearance and off-target capture that could limit EV therapy efficacy. Overall, the topic is timely and of clear interest to the readership. The manuscript is generally written in correct English and the reference list is current and largely coherent with the text. However, I recommend the follow modifications before publishing:
MAJOR REVISIONS
- Quantify EV loading efficiency: The central premise is that SIPEs encapsulate ASC-EVs and later release them to act on SMCs. The current data show a release signal (bead/CD63 flow cytometry) and uptake images, but there is no rigorous measurement of how many EVs are actually loaded per microparticle or what fraction of input EVs remain entrapped after washing. Without this, it is not possible to interpret the attenuated functional responses of SIPEs versus free EVs — these could be due to low loading rather than loss of EV potency.
- Characterize the microparticles more completely.
Only SEM and a protein ninhydrin assay are reported. Important missing descriptors are MP size distribution, surface charge, iron content and magnetic properties. These parameters affect release, biodistribution and safety. Also, a more detailed physico-chemical characterization of the microparticles both before and after EV encapsulation would substantially strengthen the manuscript by clarifying the particles’ structural integrity and their true capacity to entrap and preserve EVs.
I recommend the authors quantitatively compare particles before and after the incubation with EVs measuring size, morphology by TEM (to detect EV adsorption vs internal entrapment), surface charge and FTIR spectra (as in ref.21).
- Provide quantitative release kinetics of CD63+. The present presentation (percent CD63+ events) and single batch/semi-qualitative readout are insufficient for kinetic modelling or comparison to other systems. Report release over time (e.g., 0, 6 h, 24 h, 48 h, 72 h, 7 d).
MINOR REVISIONS
- Methods section 2.4 Fabrication of SIPEs and SIMPs:
Provide the exact EV:RSF ratio used (particles per µL or µg protein per µL), the exact volume of each component at fabrication.
- Methods 2.2 EV Isolation / Preparation and Results 3.1 EV characterization:
The manuscript uses EVs from UC, TFF and a commercial source (RoosterBio). State explicitly which experiments used which EV source (e.g., were SIPE fabrication experiments performed with in-house UC EVs or purchased EVs?). This is important because EV subpopulations and purity differ by isolation method; discuss how this may affect results.

Round 2
Reviewer 1 Report
Comments and Suggestions for Authors
No further comments
Reviewer 2 Report
Comments and Suggestions for Authors
The manuscript can be accepted in its current form.